# Review of the application of the open-source software CilOCT for semi-automatic segmentation and analysis of the ciliary muscle in OCT images

Torsten Straßer[1][*], Sandra Wagner[1], Eberhart Zrenner[1,2]

1 Centre for Ophthalmology, Institute for Ophthalmic Research, University of Tuebingen, Tuebingen, Germany, 2 Werner Reichardt Centre for Integrative Neuroscience (CIN), University of Tuebingen, Tuebingen, Germany

☯ These authors contributed equally to this work.
* torsten.strasser@uni-tuebingen.de

## Abstract

Presbyopia and myopia research shows a growing interest in ciliary muscle biometry using optical coherence tomography (OCT). Until now, segmentation of the ciliary muscle is often performed manually using either custom-developed programs or image processing software. Here we present a novel software for semi-automatic segmentation of the ciliary muscle. It provides direct import of OCT images in DICOM format, a standardized procedure for segmentation, image distortion correction, the export of anatomical ciliary muscle landmarks, like ciliary muscle apex and scleral spur, as well as a continuous thickness profile of the ciliary muscle as a novel way of analysis. All processing steps are stored as XML files, fostering documentation and reproducibility of research through the possibility of replicating the analysis. Additionally, CilOCT supports batch processing for the automated analysis of large numbers of images and the respective data export to tabulated text files based on the stored XML files. CilOCT was successfully applied in several studies and their results will be summarized in this paper.

## Introduction

Accommodation, the eye's adjustment to different viewing distances, is achieved by a contraction of the ciliary muscle, a smooth, ring-shaped muscle surrounding the crystalline lens, controlling its shape to keep the retinal image in focus. Due to a worldwide increase in myopia prevalence [1] and innovative approaches for presbyopia correction [2], research on a deeper understanding of the ciliary muscle's morphology and mechanisms has increased during the last years. Optical coherence tomography (OCT) has been frequently used to image the ciliary muscle morphology and its changes during accommodation [3–8].

To date, the shape changes of the ciliary muscle during accommodation were analyzed by measuring the ciliary muscle thickness (CMT) either in equidistant steps posterior to the

---

**Data Availability Statement:** All source code is available from GitHub (https://github.com/strator1/CilOCT).

**Funding:** SW acknowledges support from the Hector Fellow Academy (https://www.hector-fellow-academy.de/). EZ was supported by the German Research Council in the Excellence Center Program (EXC307) (https://www.dfg.de). The funders had no role in study design, data collection and analysis, decision to publish, or preparation of the manuscript.

**Competing interests:** The authors have declared that no competing interests exist.

scleral spur [3,5] or proportionally to the length of the muscle [9]. Alternatively, the change of the ciliary muscle's area during accommodation is measured [8,10,11]. Both approaches require the segmentation of the ciliary muscle, which is usually performed manually, either using built-in calipers of the OCT-system [7,8], custom-developed programs [9,12–14], or image processing software [10,11]. The diversity of tools and methods for the segmentation of the ciliary muscle as well as of the analyzed parameters results in difficulties in comparing outcomes of different studies. Furthermore, custom-developed programs used for segmentation are often not made available, which impedes the reproduction of study results.

Here we present the first software for semi-automatic segmentation and standardized analysis of the ciliary muscle, CilOCT, which is free to use and fully open-source. Biometric parameters of the ciliary muscle are automatically calculated after segmentation and image distortion correction, accounting for the different refractive indices of the tissue layers. Additionally, CilOCT performs the formation of continuous CMT profiles, which allow for the detailed comparison of the ciliary muscle morphology between different subjects or different accommodative states. CilOCT also provides an audit trail documenting all processing steps, fostering reproducibility of research. The application of CilOCT is demonstrated by summarizing the results of three of our previously published studies.

By providing this open-source software, we aim to lay the foundation for an improved comparability of studies in the field of accommodation and ciliary muscle research using OCT, thereby fostering a better understanding of both anatomy and functionality of the ciliary muscle and its role in myopia and presbyopia development.

## Material and methods

### Software architecture

CilOCT was developed as a Java (Version 8) application with a graphical user interface based on JavaFX and uses several open-source libraries. The user interface leverages the ribbon-paradigm, a graphical control element consisting of a set of toolbars placed on several tabs, first introduced in Microsoft Office 2007, using the open-source library FXRibbon (version 1.0.1, BSD license) to simplify and accelerate user interaction [15]. The image import makes use of the standard Java ImageIO classes for common image file formats (BMP, JPEG, PNG) and dcm4che-imageio (version 3.3.8, MPL 1.1 license) for raw DICOM files of the Zeiss Visante™ anterior segment-OCT (AS-OCT). Mathematical calculations, especially matrix algebra and polynomial spline fitting, are implemented using the Apache commons-math3 library (version 3.6.1, Apache license 2.0). Im- and export of guiding landmark coordinates to XML leverages jackson-databind (version 2.9.7, Apache license 2.0). CilOCT uses Apache Maven 3 as project management and build-automation system [16].

### Software functionalities

CilOCT enables the semi-automatic segmentation of the ciliary muscle in images acquired using OCT. It allows OCT images to be imported from raw DICOM images of the Zeiss Visante™ AS-OCT [17] or common image formats (i.e. BMP, JPEG, PNG). DICOM images of the Zeiss Visante™ AS-OCT are automatically resized and rotated for the correct display [14]. The software can easily be extended for other OCT instruments. After importing the image, its contrast can either be manually increased or decreased or adjusted by adaptive histogram equalization [18] to enhance the edges of the ciliary muscle for easier placement of guiding landmarks along the muscle's boundaries. Changing the contrast does not affect the following automatic segmentation. The semi-automatic segmentation using the manually placed guiding landmarks results in polynomial spline fits of the borders of the ciliary muscle, the anterior

chamber, the scleral-air boundary, as well as the surface of a possibly used contact lens. The coordinates of the guiding landmarks, along with any processing performed to enhance image quality, can be stored as an XML file, which allows for reliable reproduction of the segmentation results. Before the determination of ciliary muscle biometric parameters, the image distortion resulting from the refraction of the infrared light beam on boundaries with different refractive indices is corrected to enable the calculation of the true layer thicknesses. Image distortion correction is performed and applied to the segmentation using ray-tracing based on Snell's Law and the refractive indices of the respective layers (sclera: 1.41, ciliary muscle: 1.38, aqueous fluid: 1.34 [19]). The values of the refractive indices of the different layers are configurable. Finally, based on the distortion corrected polynomial splines, the following biometric parameters are calculated: ciliary muscle apex, scleral spur, perpendicular axis (a selective anterior CMT reading taken at the apex), ciliary muscle apex shift, ciliary muscle area, and CMT profile [20]. The determined parameters can be copied to the system clipboard or saved as character separated text files. The batch-processing feature allows for the analysis of large numbers of images based on previously created and stored XML segmentation files and to subsequently export the anatomical ciliary muscle landmarks, determined parameters, and CMT profiles as tabulated text files.

## Imaging of the ciliary muscle using OCT

In our previous studies, a Zeiss Visante™ AS-OCT (Carl Zeiss Meditec AG, Jena, Germany; enhanced high-resolution corneal mode, 512 A-scans, scanning size: 10 mm x 3 mm, wavelength 1310 nm) was used. To image the ciliary muscle, subjects were positioned in front of the OCT device with their head placed in the combined head and chin rest. After adjusting the imaged area using pupil center alignment, the subjects were instructed to shift their gaze about 40˚ to an external target without moving their head but keeping it parallel to the instrument. This allowed imaging the entire ciliary muscle as shown in Fig 1. A detailed description of the process including a sketch of the experimental setup can be found in [20].

## Segmentation of the ciliary muscle using CilOCT

The ciliary muscle segmentation consists of a two-stage process:

1. Guiding landmarks (about 10–15 per boundary) are placed manually roughly along the borders between (1) sclera and muscle, (2) muscle and pigmented epithelium, as well as (3) along the muscle boundary towards the iridocorneal angle (Fig 2b). For each of the borders, a polynomial spline is fitted through the coordinates of these landmarks.

2. For each pixel of the fitted polynomial splines, the coordinates of the maximum brightness gradient within a distance of ± 5 pixels perpendicular to the spline are determined (Fig 2c) and used for a second, exact polynomial spline fit (Fig 2d).

In addition to the ciliary muscle borders, the scleral-air boundary, as well as the borders of the anterior chamber are segmented by a simple 8-neighbors flood-fill algorithm using a manually placed starting point (purple landmark in Fig 2b, turquoise area in Fig 2c, orange and yellow borders in Fig 2d). These borders are required for the following distortion correction using Snell's Law and the refractive indices of the different tissues (infrared light rays depicted as yellow lines perpendicular to the upper border of the image and as red/orange lines after refraction at the different tissue borders in Fig 2e). Similarly, a possibly worn contact lens can be segmented (see S1 Movie for an example). The thresholds of the flood-fill algorithm can be manually adjusted for the different regions. Fig 2f depicts the determined biometric

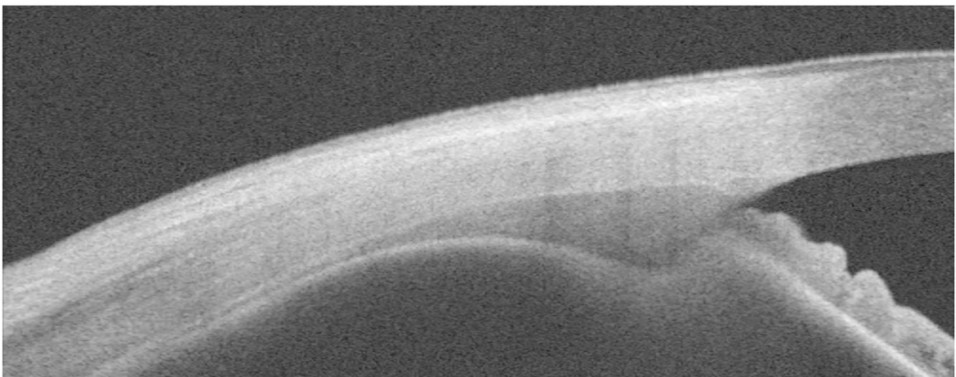

**Fig 1. OCT image of the temporal ciliary muscle of the right eye.** The ciliary muscle appears as a dark shaded bent triangular structure below the sclera. The image was acquired using a Zeiss VisanteTM anterior segment-OCT, covering a size of 10 mm length and 3 mm tissue depth.

parameters scleral spur, ciliary muscle apex, perpendicular axis, and ciliary muscle area after distortion correction.

In addition to the biometric parameters ciliary muscle apex, scleral spur, perpendicular axis, ciliary muscle apex shift, and ciliary muscle area, the CMT profile is calculated as the distance between the intersections of a secant perpendicular to the scleral-conjunctival surface with the scleral-muscle border and the muscle-pigmented epithelium border, respectively (Fig 3). This approach is used to normalize different shapes of the scleral-conjunctival surface [20].

A demonstration of the segmentation process using the open-source software CilOCT is available as a movie in the supporting information (S1 Movie). To test the segmentation of the ciliary muscle using the CilOCT software, the OCT image file in DICOM format (S1 File) along with the corresponding guiding landmarks file (S2 File) can be used.

### Application of CilOCT in previous studies

Images and results reported here were collected in previously published studies [20–22] in which CilOCT was applied. The studies followed the tenets of the declaration of Helsinki and were approved by the Institutional Review Board of the medical faculty of the University of Tuebingen (376/2017BO2).

### Results

In a first study, the reliability of the ciliary muscle segmentation performed with CilOCT was evaluated by analyzing the OCT images of fifteen near-emmetropic volunteers. Intra- and inter-examiner as well as intra-session repeatability using the two parameters ciliary muscle area (CMA) and perpendicular axis (PA) (Fig 3) were assessed [20]. Six OCT images of the right temporal ciliary muscle were taken in each of the volunteers during both, near and far accommodation. A short break followed by subject re-alignment after three images in both accommodation states, respectively, allowed for the intra-session repeatability analysis. The ciliary muscle segmentation was performed twice by the same examiner (SW) within one week to assess intra-examiner repeatability and additionally by a second examiner (TS) once to evaluate inter-examiner repeatability. The intra-examiner analysis of the CMA revealed a statistically significant mean difference of 0.06 mm$^2$ between the two successively performed segmentations, probably caused by a training effect of the examiner [20]. However, the analysis of the PA showed that the resulting mean difference between the two segmentations was

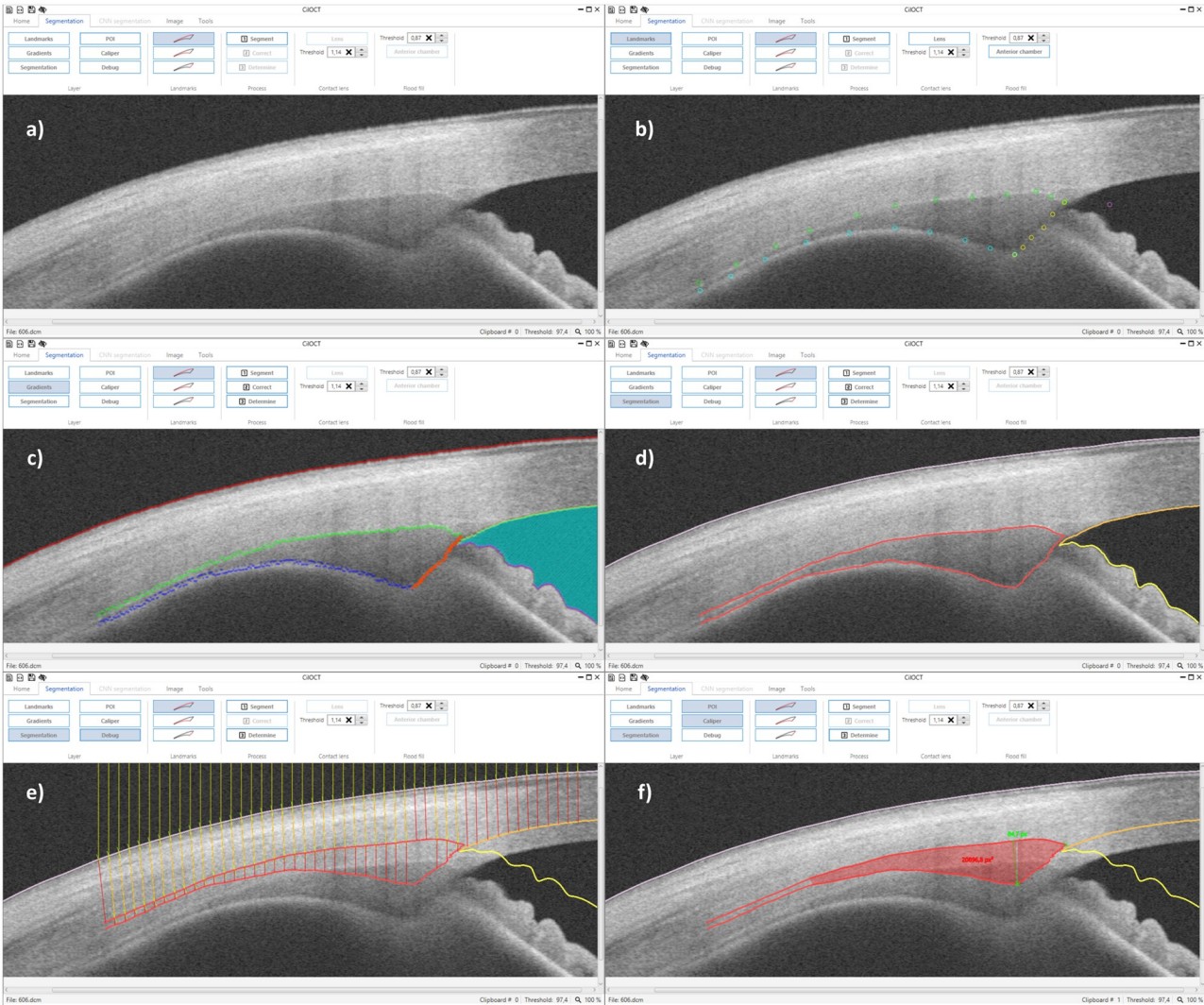

**Fig 2.** An illustrative example of the segmentation of the ciliary muscle: (a) The imported raw DICOM OCT image. (b) Manually placed guiding landmarks for the upper (green circles), lower (blue circles), and the nasal or temporal border (yellow circles) of the ciliary muscle, and a landmark in the iridocorneal angle (purple circle) as starting point of the 8-neighbor flood-fill algorithm (guiding landmarks size is enlarged for better visibility). (c) Maximum brightness gradients determined from the first rough polynomial spline fit of the landmarks. (d) Polynomial splines of the final segmentation. (e) Distortion correction based on Snell's Law using the refractive indices of the different tissue layers. The incoming light rays of the infrared light beam and how they are refracted at the boundary between air and scleral-conjunctival tissue are exemplified as yellow lines perpendicular to the top of the images. Red and orange lines depict the refracted light rays. (f) Distortion corrected segmentation and determined biometric parameters scleral spur, ciliary muscle apex, perpendicular axis, and ciliary muscle area (all values in pixels).

22.97 μm (≙ 2.94 px) and therefore within the range of the tomographer's resolution [20]. The intra-session repeatability, analyzed using the mean value of the CMA of the first and the last three images segmented by one examiner, was found to be good with an ICC of 0.88 and a paired t-test revealed no statistically significant difference [20]. A comparison between the CMA derived from the segmentation of the two examiners showed no statistically significant difference and inter-examiner repeatability was found to be good with an ICC of 0.87 [20].

The analysis of the CMT profiles as a novel parameter resulted in the identification of regions of a posterior thickening and an anterior thinning of the ciliary muscle when accommodating

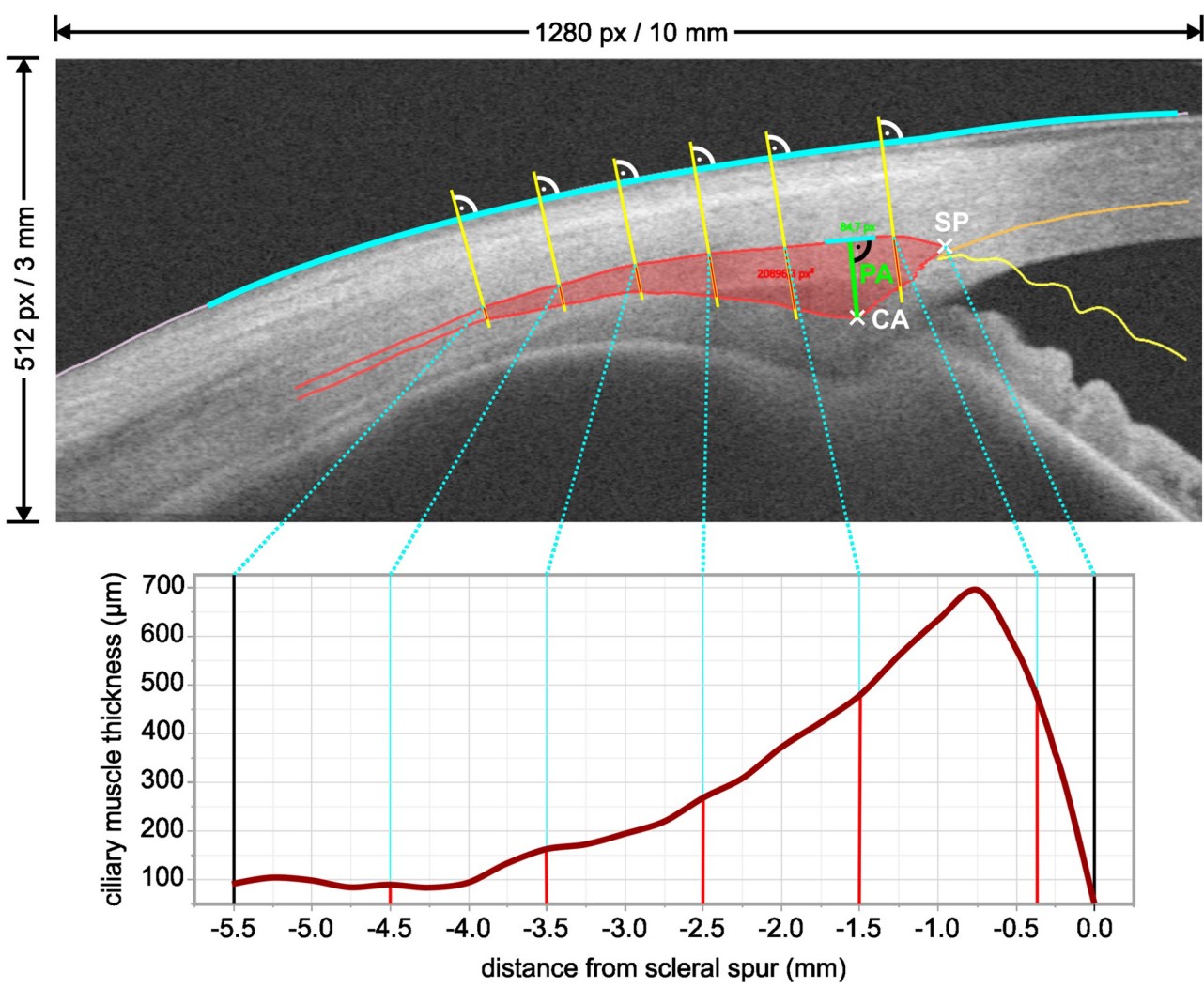

**Fig 3. Ciliary muscle thickness profile.** The profile is calculated as the distance between the intersection of a secant perpendicular to the scleral-conjunctival surface with the scleral-muscle border and the muscle-pigmented epithelium border (red lines).

by 3 D. This could not be observed in such detail using selective thickness measurements based on equidistant or proportional steps as done before [20].

Referring to the accumulating evidence of an association between near vision and myopic development, the second study using CilOCT aimed at comparing the accommodation process of emmetropic (n = 18) and myopic (n = 20) young adults regarding both refractive power changes of the crystalline lens and CMT changes for different accommodation demands. The OCT image analysis revealed that accommodation mechanics of the two refractive groups differ significantly with respect to the ciliary muscle's shape, movement, and thickness changes. The myopic ciliary muscle was found to be thinner than the emmetropic one in the anterior region, but thicker posteriorly. Moreover, myopes exhibited smaller anterior thickness changes of the ciliary muscle during accommodation, but they showed a larger muscle movement compared to their emmetropic counterparts. On the contrary, refractive power changes of the crystalline lens during a dynamic target presentation of the pattern far-near-far were not found to be statistically significantly different between emmetropes and myopes [21].

In this context, a third study was performed to investigate the effect of prolonged nearwork on the ciliary muscle and the refractive power changes of the lens. OCT imaging of the ciliary muscle as well as photorefraction recordings of accommodation for a demand of 4 D were performed in 17 emmetropes and 18 myopes before and after a continuous 30-min reading task at 25 cm distance. The analysis of the changes of the CMT profiles revealed an unexpected thinning of the muscle after the nearwork task in both refractive groups. However, only myopes showed a sustained relative myopic shift of their lens power after the prolonged close work, while it remained rather stable in emmetropes [22].

These previous studies using CilOCT gave new insights into the myopic ciliary muscle's anatomical structure and reactions during regular and sustained accommodation and the respective differences to the emmetropic muscle. The findings could therefore be instrumental in the further assessment of a possible role of the ciliary muscle in myopigenesis.

## CilOCT availability

The source code of CilOCT is available in a GitHub repository at https://github.com/strator1/ciloct and is licensed under the open-source GNU General Public License 3 (GNU GPLv3) to ensure that any derivative work is kept as open-source.

A pre-built version of the software can be downloaded from https://github.com/strator1/CilOCT/releases/tag/2.0.6. The software runs on any operating system with compatible Java Runtime Environment (JRE, version $\geq$ 8) installed (i.a. Windows, macOS, Linux).

## Discussion

The open-source software CilOCT allows for a semi-automatic segmentation and a standardized analysis of the ciliary muscle with an audit trail documenting the processing steps for reproducible results. Reproducible research requires, as stated in a recent article, "Making computer codes available to others together with a detailed log of every action" in order to "provide[s] a level of detail greater than experimental descriptions using natural language" [23]. A minimum requirement is, in addition to the study protocol and the description of the measurement procedures, the data with descriptive metadata as well as the analysis software and, in case the original software is not supported anymore, the source code [24]. By providing a detailed log of the image processing and segmentation steps, CilOCT opens up the possibility of a reliable replication of the ciliary muscle's segmentation and subsequent analysis. CilOCT's batch feature enables time- and resource-saving automatic and reproducible determination of biometric parameters of the ciliary muscle in large numbers of OCT images resulting from longitudinal studies or those involving large cohorts. Such studies will help to answer questions regarding the relationship between the ciliary muscle morphology and the accommodation dynamics, and the possible involvement of the ciliary muscle in the development of myopia. In case a relationship between myopia progression and changes in the ciliary muscle morphology was found, the here presented anatomical measures might serve as additional endpoints in interventional studies aiming at myopia control. Similarly, biometric parameters of the ciliary muscle derived from the application of OCT imaging may help to answer the still open question of the ciliary muscle's contribution to the development of presbyopia. While it is well accepted that a progressive hardening of the crystalline lens finally results in presbyopia [25], it is unclear if and to which amount the ciliary muscle is active at that stage [26], though several studies found indications for a preserved functionality well beyond the onset of presbyopia [27–30]. The analysis of the CMT profiles may add further evidence to a purely lenticular-based theory of presbyopia.

Furthermore, CilOCT's unique use of a coordinate system with the scleral spur as the point of origin and the intersections between upper ciliary muscle border and perpendiculars to the sclera as basis for the abscissa renders the analysis independent of the ciliary muscle's position and orientation within the OCT image [20]. CMT profiles of subjects with different refractive errors, of different ages, or in different accommodative states can thereby be accurately overlaid and compared. In addition to the commonly used ciliary muscle area measurements and discrete CMT readings, CMT profiles offer a novel analysis method for visualization and measurement of fine details of the thickness changes, which might remain undetected using current analysis techniques.

By providing CilOCT as free and open-source software, we aim to establish a better comparability of studies assessing the ciliary muscle's structure, features, and changes during accommodation, thereby bringing forward both, myopia and presbyopia research.

## Conclusions

In this paper, we presented CilOCT, an open-source software for the semi-automatic segmentation of the ciliary muscle in OCT images. It provides automatic calculation of several biometric parameters for analysis of the ciliary muscle's morphology as well as a novel approach for the creation of a continuous ciliary muscle thickness profile. The software is written in Java and therefore runs on Windows, macOS, and Linux/Unix. CilOCT can import OCT images from either the raw DICOM format of the Zeiss Visante™ AS-OCT or standard image files. Further sources can easily be added. CilOCT performs an image distortion correction accounting for the difference in the refractive index of the anatomical layers. All image processing steps and the guiding landmarks can be stored as an XML file, allowing for a reproducible segmentation and an automated batch processing. CilOCT was successfully employed in several studies and we believe that it can foster further research on the structure and the function of the ciliary muscle in myopia and presbyopia.

## Supporting information

**S1 Movie. Demonstration of the segmentation of the ciliary muscle in an OCT image using the open-source software CilOCT.**
(MP4)

**S1 File. Example OCT image of the ciliary muscle of one of the authors (TS).**
(DCM)

**S2 File. Saved guiding landmarks file for the segmentation of the ciliary muscle in the OCT image (S1 File).**
(XML)

## Acknowledgments

We thank Marco Ruggeri for advice regarding the analysis software and Anne Kurtenbach for proofreading the manuscript.

## Author Contributions

**Conceptualization:** Sandra Wagner, Eberhart Zrenner.

**Investigation:** Sandra Wagner.

**Methodology:** Torsten Straßer, Sandra Wagner.

**Project administration:** Eberhart Zrenner.

**Resources:** Eberhart Zrenner.

**Software:** Torsten Straßer.

**Supervision:** Torsten Straßer, Eberhart Zrenner.

**Validation:** Sandra Wagner, Eberhart Zrenner.

**Writing – original draft:** Torsten Straßer, Sandra Wagner.

**Writing – review & editing:** Torsten Straßer, Sandra Wagner, Eberhart Zrenner.

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
