## [Decision Letter · Decision Letter 0]

18 Mar 2020

PONE-D-20-01371

CilOCT: An open-source software for semi-automatic segmentation and analysis of the ciliary muscle in OCT images

PLOS ONE

Dear Dr. Straßer,

Thank you for submitting your manuscript to PLOS ONE. After careful consideration, we feel that it has merit but does not fully meet PLOS ONE’s publication criteria as it currently stands. Therefore, we invite you to submit a revised version of the manuscript that addresses the points raised during the review process.

We would appreciate receiving your revised manuscript by Apr 30 2020 11:59PM. To enhance the reproducibility of your results, we recommend that if applicable you deposit your laboratory protocols in protocols.io, where a protocol can be assigned its own identifier (DOI) such that it can be cited independently in the future. For instructions see: http://journals.plos.org/plosone/s/submission-guidelines#loc-laboratory-protocols

We look forward to receiving your revised manuscript.

Kind regards,

Asaf Achiron

Academic Editor

PLOS ONE

Journal Requirements:

2. We strongly encourage authors to avoid using figures or tables that have already been published, because this can cause serious copyright concerns. We recommend that you replace figure 3 with a reference to the previous publication.

If you feel that it must be included in the current submission, please confirm that you have express written permission from the copyright holder of figure 3 to publish it under the CCAL license.

Anything published under this license is available online to read, download, copy, distribute, and use in any way (with attribution) without further permission. Please see http://www.PLOSone.org/static/guidelines.action#preparation with reference to the paragraph titled Figures for more information. You must ensure that the permission from the copyright holder specifically allows you to publish the content under the CCAL license.

To seek permission, we recommend that you contact the original copyright holder with the following text: “I request permission for the open-access journal PLOS ONE to publish XXX under the Creative Commons Attribution License (CCAL) CC BY 4.0 (http://creativecommons.org/licenses/by/4.0/us/). Please be aware that this license allows unrestricted use and distribution, even commercially, by third parties. Please reply and provide explicit written permission to publish XXX under a CC BY license.”

Please upload the granted permission to the manuscript as a supporting information file. In the figure caption of the copyrighted figure, please include the following text: “Reprinted from [ref] under a CC BY license, with permission from [name of publisher], original copyright [original copyright year].”

If you are unable to obtain permission from the original copyright holder, please either i) remove the figure or ii) supply a replacement figure, which can be for illustrative purposes only. Please check copyright information on all replacement figures. If applicable, please specify in the figure caption text when a figure is similar but not identical to the original image, and is therefore for illustrative purposes only.

3. Your ethics statement must appear in the Methods section of your manuscript. If your ethics statement is written in any section besides the Methods, please move it to the Methods section and delete it from any other section. Please also ensure that your ethics statement is included in your manuscript, as the ethics section of your online submission will not be published alongside your manuscript.

Reviewers' comments:

Reviewer's Responses to Questions

**Comments to the Author**

1. Is the manuscript technically sound, and do the data support the conclusions?

Reviewer #1: Yes

Reviewer #2: No

Reviewer #3: No

2. Has the statistical analysis been performed appropriately and rigorously? 

Reviewer #1: Yes

Reviewer #2: No

Reviewer #3: N/A

3. Have the authors made all data underlying the findings in their manuscript fully available?

Reviewer #1: Yes

Reviewer #2: No

Reviewer #3: Yes

4. Is the manuscript presented in an intelligible fashion and written in standard English?

Reviewer #1: Yes

Reviewer #2: No

Reviewer #3: Yes

5. Review Comments to the Author

Reviewer #1: Thank you for allowing me to review this interesting paper.

Research article-is the very suitable for this article.

Open-Source is the most talked issue in programing in the last years It’s very good to see it in our field.

Remarks

abstract

1. I think you need to elaborate more about the “batch processing”- Most of the readers don’t know enough about it.

Intro

2. You are talking about different refractive Indices please elaborate more including the numbers you are talking about.

3. You’re explaining how you took the images- Its not very clear- please add a photo how you’ve taken it. Moreover- you’re siting [16] your own article.

Method

4. When you are talking about figure 2e and 2f is not very clear. It needs to be more elaborated.

5. The colors in 2b are not sharp enough- very difficult to see it and understand.

6. In line 126 you are talking about “distortion correction”- it is not clear enough since you’ve already talked about the ciliary muscle in 2d.

Discussion

7. You are talking about 6 OCT images in 15 Volunteers – why is the discrepancy?

I think it is a very interesting research article with a very updated topic. Though with a small number of subjects.

Reviewer #2: editor comments

General

1 was intrigue by the title of this study – to evaluate the ciliary muscles – indeed a great topic as accommodation presbyopia and myopia our daily life.

2 The ability to standardized reports of imaging is of benefit to all researchers as it create a common language for definition and classification

cover letter

1 I understand you offer the code for the program. Can you offer the software itself for those who do not have coding experience? An online tool will be great

Intro

1 some font and formatting could be improved- please go over and make sure all is correct= for example reference 15 is not in the same font

2 FIG 1- please refer the reader to the figure from the text.

3 “CilOCT supports the import of images of the ciliary muscle either from standard image format 54 (BMP, JPEG, PNG) or from raw Zeiss DICOM files [15]. In the case of an import from raw Zeiss 55 DICOM format, the images are automatically rotated and resized [14].”

and

“To image the ciliary muscle subjects were positioned in front of an anterior segment-OCT (VisanteTM 62 AS-OCT, Carl Zeiss Meditec AG, Jena, Germany; enhanced high-resolution corneal 63 mode, 512 A-scans, scanning size: 10 mm x 3 mm, wavelength 1310 nm) with their head placed 64 in a combined head and chin rest [16]. Since the infrared beam can only penetrate the sclera but 4 65 not the highly pigmented iris [17], the subjects were instructed to shift their gaze about 40° to an 66 external target while keeping their head parallel to the instrument. This allowed imaging the entire 67 ciliary muscle as shown in Fig 1 [16]. Images and results reported were collected in previously 68 published studies [16,18,19]. The studies followed the tenets of the declaration of Helsinki and 69 were approved by the Institutional Review Board of the medical faculty of the University of 70 Tuebingen (376/2017BO2)”

should move to the methods

4 The intro lacking the purpose of this study: please state clearly that this is a review study aiming to introduce the software- no new data is given her- this is an overview report

you could add that to the title that it is a review

methods-

1 CilOCT is developed- “was”

2 The user interface leverages the ribbon-paradigm to simplify and accelerate user interaction- please explain in words that average human can explain that is this ribbon? give example or photo?

3 Table 1 – please refer the reader from the text- I think this could be incorporated in the text actually as the table does not add much

4 ccontrast can either be manually increased or decreased or adjusted by adaptive histogram equalization-Is different contrast affect the measurements?

5 Illustrative Examples – could you provide capture screen video as a tutorial? ah S1 movie Ok

6 is a windows version exist that a regulat user can download and install and work?

Discussion

1 no results sections?

for example, from 100 scans how many were accurate? what is the false positive negative? timing of the procedure? etc.

2 The intra-examiner analysis of the CMA revealed---where is this reported? how many examiners? how many test? what was the statistical test

please follow a intro- method-results-conclusion design- we are all used to it

ciliary muscle thickness profiles- you should show that this is indeed an helpful parameter

conclusion- after reading the study- I must admit that the format or order of the study made me confused- what was the propose of this study? if 3 previous studies were already puclished what is new in the current one- if it a review please descrivvve in depth the previous studies (n, methods etc) summeried the results etc.

I think if the aim is to present a new software the the format should be technical and present data that show that the software is more accurate them manul segmention that it reduce cost and time etc-

Reviewer #3: This manuscript summarizes the authors’ experience with a software for the analysis of OCT images of the ciliary muscle. This experience is based on 3 of the authors’ previous studies (ref 16, 18 and 19). The paper is well written, and demonstration of the technique is clear and appears to be easy to reproduce. However, in my opinion, the authors did not provide adequate justification to publish this manuscript, since there is no evidence of any new original research compared to the 3 previous reports. This manuscript may benefit from being presented as a review.

At list one of the figures was published in one of the authors’ previous studies, though this was adapted with permission. This seems inappropriate to me.

6. PLOS authors have the option to publish the peer review history of their article (what does this mean?). If published, this will include your full peer review and any attached files.

Reviewer #1: Yes: Evgeny Gelman

Reviewer #2: Yes: Asaf Achiron

Reviewer #3: No

---

## [Author Response · Author response to Decision Letter 0]

30 Apr 2020

We ensured that the manuscript meets PLOS ONE's style requirements, including those for file naming.

We supplied a replacement figure for figure 3

We added the ethics statement to the Methods and material section.

Please find all other responses to reviewer questions and comments in the submitted document "Response to Reviewers.docx"

---

## [Decision Letter · Decision Letter 1]

26 May 2020

Review of the application of the open-source software CilOCT for semi-automatic segmentation and analysis of the ciliary muscle in OCT images

PONE-D-20-01371R1

Dear Dr. Straßer,

We are pleased to inform you that your manuscript has been judged scientifically suitable for publication and will be formally accepted for publication once it complies with all outstanding technical requirements.

With kind regards,

Asaf Achiron

Academic Editor

PLOS ONE

Additional Editor Comments (optional):

Reviewers' comments:

Reviewer's Responses to Questions

**Comments to the Author**

1. If the authors have adequately addressed your comments raised in a previous round of review and you feel that this manuscript is now acceptable for publication, you may indicate that here to bypass the “Comments to the Author” section, enter your conflict of interest statement in the “Confidential to Editor” section, and submit your "Accept" recommendation.

Reviewer #2: All comments have been addressed

Reviewer #3: All comments have been addressed

2. Is the manuscript technically sound, and do the data support the conclusions?

Reviewer #2: Yes

Reviewer #3: Yes

3. Has the statistical analysis been performed appropriately and rigorously? 

Reviewer #2: N/A

Reviewer #3: N/A

4. Have the authors made all data underlying the findings in their manuscript fully available?

Reviewer #2: Yes

Reviewer #3: Yes

5. Is the manuscript presented in an intelligible fashion and written in standard English?

Reviewer #2: Yes

Reviewer #3: Yes

6. Review Comments to the Author

Reviewer #2: I think this will be a good tool.

the author did correct all my remarks

I have no more comments

good luck

Reviewer #3: Thank you for the request to re-review this article.

The authors addressed and amended all issues in a satisfactory manner, most importantly changing the format of the paper to a review. No further comments. My recommendation is to accept this valuable manuscript.

7. PLOS authors have the option to publish the peer review history of their article (what does this mean?). If published, this will include your full peer review and any attached files.

Reviewer #2: No

Reviewer #3: Yes: Tal Koval

---

## [Editor Report · Acceptance letter]

28 May 2020

PONE-D-20-01371R1 

Review of the application of the open-source software CilOCT for semi-automatic segmentation and analysis of the ciliary muscle in OCT images 

Dear Dr. Straßer:

I am pleased to inform you that your manuscript has been deemed suitable for publication in PLOS ONE. Congratulations! Your manuscript is now with our production department. 

With kind regards,

on behalf of

Dr. Asaf Achiron 

Academic Editor

PLOS ONE